# A Wrong Fate Decision in Adipose Stem Cells upon Obesity

**DOI:** 10.3390/cells12040662

**Published:** 2023-02-19

**Authors:** Yiu-Ming Cheung, Chui-Yiu-Bamboo Chook, Hoi-Wa Yeung, Fung-Ping Leung, Wing-Tak Wong

**Affiliations:** 1School of Life Sciences, The Chinese University of Hong Kong, Hong Kong, China; 2State Key Laboratory of Agrobiotechnology, The Chinese University of Hong Kong, Hong Kong, China

**Keywords:** stem cell aging, inflammation, hypertrophic obesity, adipogenesis, fate determination

## Abstract

Progress has been made in identifying stem cell aging as a pathological manifestation of a variety of diseases, including obesity. Adipose stem cells (ASCs) play a core role in adipocyte turnover, which maintains tissue homeostasis. Given aberrant lineage determination as a feature of stem cell aging, failure in adipogenesis is a culprit of adipose hypertrophy, resulting in adiposopathy and related complications. In this review, we elucidate how ASC fails in entering adipogenic lineage, with a specific focus on extracellular signaling pathways, epigenetic drift, metabolic reprogramming, and mechanical stretch. Nonetheless, such detrimental alternations can be reversed by guiding ASCs towards adipogenesis. Considering the pathological role of ASC aging in obesity, targeting adipogenesis as an anti-obesity treatment will be a key area of future research, and a strategy to rejuvenate tissue stem cell will be capable of alleviating metabolic syndrome.

## 1. Introduction

Adipose tissue (AT) is undoubtedly a pleiotropic organ in mammals, distributing in different anatomical depots over the body [1]. Dysregulation in body adiposity, either insufficiency or excess, is often associated with lipodystrophy or obesity, respectively. Although AT was solely thought to be a lipid-storage tissue in the very beginning, advancements in biochemical analysis and transgenic technology have deepened its endocrine role in energy homeostasis. Given its overwhelming plasticity and dynamicity, AT is predominantly made up of adipocytes, accounting for more than 90% of the cellular volume of AT owing to their single large lipid droplet [2]. The remaining cell types can be simply divided into neuron and stromal vascular fraction (SVF) cells, which include endothelial cells, fibroblasts, leukocytes, preadipocyte, and adipose stem cells (ASCs). Lining in perivascular locations and the reticular interstitium, ASCs (also known as adipose stromal/progenitor cells) are regarded as a kind of mesenchymal stem cells (MSCs), giving rise to self-renewal and adipogenic differentiation [3,4,5].

Looking back to the 1960s, the identification of hematopoietic stem cells (HSCs) helped conceptualization of the modern stem cell hypothesis, especially its functional relationship with their microenvironment [6]. Thereafter, understanding in stem cell concept and phenotyping gained wide popularity, but the discovery of postnatal adult stem cells was only limited to bone marrow MSCs as well as neural stem cells before the millennium [7,8]. Stepping into the 21st century, a study performed by Zuk et al. was first to have identified an unspecialized cell population with differentiation capacity and clonogenicity, termed processed lipoaspirated (PLA) cells, thereby considering a true stem cell population among AT [9,10]. Around the same time, Gimble’s team also identified the multipotency of adipose-derived stromal cells towards adipogenic or osteogenic differentiation [11]. Since then, despite inconsistent ASC definitions from different studies, the ASC discovery undoubtedly brought enormous potentials into tissue engineering and adipogenesis-associated research [12].

Adipocyte turnover is a dynamic process that replaces around 10% of total adipocytes in human annually [13]. As an origin of adipocytes, ASC homeostasis is particularly important, and it is closely conditioned by the adiponiche, the microenvironment where the ASCs reside and interact with other cells. Nevertheless, the pathological alterations in ASCs determine the adipogenic profiling which deteriorates AT dysregulation. Along with the increased nutrient-energy stress, cellular aging is accelerated across different species [14,15]. Owing to high levels of lipotoxic factors and oxidative stress, increased cellular senescence is a feature of AT. While major understanding about stem cell aging refers to the cell cycle arrest and release of senescence-associated secretory proteins (SASPs), the unique characteristics of stem cells such as lineage commitment are susceptible to cellular reprogramming in a defective niche [16]. Of note, a failure to undergo adipogenesis will inevitably result in hypertrophy as a fundamental cause of adult obesity, thereby triggering other metabolic complications and ultimately premature death [15,17]. In this review, we will discuss how ASCs are dysregulated, particularly in white ATs, unless specified, with a focus on lineage commitment in obesity. Dissecting the role of ASC in the adiposopathy may explore an avenue targeting aberrant fate determination to restore their stemness.

## 2. Immunophenotyping and Cellular Heterogeneity

In mammals, the paradigm of AT types is based on their molecular, morphological, and functional characteristics, by which they can be simply classified into white AT (WAT), brown AT (BAT), and beige AT (Table 1). In all kinds of AT, de novo adipogenesis from ASC is crucial to maintain tissue homeostasis and determine adiposity [13]. Indeed, a set of minimal criteria for defining ASCs has been proposed by the International Society for Cellular Therapy in 2013 for the sake of providing a uniform characterization for scientific discussions among investigators. ASC is considered as CD10^+^ CD13^+^ CD36^+^ CD73^+^ CD90^+^ CD105^+^ CD45^−^ cell with 2–30% CD34 and CD106 expression [18]. Beyond the definition, depletion of CD31, CD235a, and Lin are also often examined via fluorescence-activated cell sorting (FACS).

In fact, the immunophenotypes of ASC reported by studies vary between different anatomical depots, sorting strategies, species, and disease status. One example is the depot-specific characteristics observed in the ASCs derived from WAT. While the ASCs from subcutaneous fat specifically express CD10, with a greater adipogenic capacity, those from visceral regions show an upregulated CD200 expression [23]. Aside from these features, the varied expressions of other proteins observed among the entire ASC pool also lead to questions of how heterogenous ASCs truly are and what physiological roles they play. For instance, the expressions of the mesenchymal markers CD24, CD29, CD34, and PDGFRα/β were suggested to be associated with the stemness and adipogenic capacity of ASCs [4,24,25,26,27,28].

In recent years, the advancements in single-cell RNA-sequencing (scRNA-seq) further distinguished the adipose stem and progenitor cell population into CD55 ^+^ DPP4^+^ early progenitor with stem-like properties, which is functionally and phenotypically distinct from VAP ^+^ ICAM1^+^ committed progenitor and CD142 ^+^ adipogenesis regulator population [5]. Nevertheless, not all the ASCs are responsible for driving the adipocyte turnover. At a lean condition, CXCL14^+^ population inhibits the macrophage infiltration into inguinal WATs, whereas the SDC ^+^ ASCs promote the fibrotic remodeling and inflammation upon obesogenic challenge [29]. Similarly, another stem cell subset enriched with CD9 in visceral WATs also favors the fibrogenesis during obesity [30]. Of note, PDGFRβ- and LY6C-expressing fibro-inflammatory progenitor was identified recently; it lacks adipogenic capacity and instead displays pro-fibrogenic and pro-inflammatory gene programs which likely contribute to macrophage infiltration into AT upon obesity [31,32,33]. Going forward, in relation to development and pathological alterations, determining the functional significance of these distinct subsets will be the next step.

## 3. ASC Renewal

Stemness and multipotency, which determine the lineage commitment and differentiation of stem cells, are essential to tissue homeostasis. Through a tight network of regulation, the self-renewal program of stem cells is initiated to drive cell proliferation, maintain its multipotency, as well as avoid exhaustion or senescence [34]. Mechanistically, the cell cycle progression is limited by several factors to avoid overproliferation or prematurity. It is driven by the activities of cyclin and cyclin-dependent kinases (CDKs), whereas it is negatively regulated by CDK inhibitors (CDKi) to prevent premature division. Similar to other adult stem cells, ASCs are often quiescent when residing in G_0_ phase. Upon the stimulation by mitogen, the complexes assembled by cyclins and CDKs, such as cyclin E-CDK2 and cyclin D-CDK4/6, promote cell cycle re-entry from quiescence to proliferation by phosphorylating and thus inactivating the Rb proteins, which in turn stimulates the E2F transcription factors to induce DNA replication (Figure 1) [35].

In fact, the quiescent state offers the advantages of enhancing stress resistance, preserving genome integrity, and increasing their stemness capacity [34]. The G_0_/G_1_ fraction is encouraged by the upregulated expression of CDKi, of which these kinase inhibitors are constituted by Ink4 and members from the Cip/Kip families (Figure 1). Among the quiescent DLK^−^ CD34^+^ ASC population from human donors, the anti-proliferative CDKi, such as p21^Cip1^, p27^Kip1^ and p57^Kip2^, are highly expressed, while the expression of cyclins and CDKs is undetected [21]. Another study on human ASCs further demonstrated that p57^Kip2^ is responsible for inducing cellular quiescence and cycle exit through the downregulation of the CDK2-cyclinE1 complex [36]. Moreover, the expression of stemness markers KLF4 and c-MYC is also shown to be enhanced in quiescent cells. While KLF4 prevents overproliferation by inducing CDKi expressions, c-MYC serves as a negative feedback loop against KLF4 by promoting cell cycle progression [21]. However, the paradoxical observation of whether ASCs dynamically interconvert between quiescence and proliferation by balancing KLF4 and c-MYC awaits investigation.

Apart from the self-regulatory pathways, the cell cycle machinery is also dependent on environmental factors. PDGFα signaling, which was previously shown to be involved in the self-renewal in glioblastoma cancer stem cell, has a great influence on both the self-renewal and proliferative capacities of dermal CD24 ^+^ ASCs via the PI3K/AKT pathway [37,38].

## 4. Aberrant Fate Determination as ASC Aging

Stem cell aging is an inevitable process over the life course, in which it plays a role in organismal aging. Given the interdependent and reciprocal relationship between stem cells and their niche, nutrient overload drives the ASC aging in the adiponiche where proinflammatory factors and reactive oxygen species are highly expressed. It is noted that the ASCs derived from obese patients display a classical signature of upregulated CDKi such as p16 and p53, as well as increased SASP factors such as IL6 and MCP1, promoting inflammation in the microenvironment [39,40]. Even through the ASC population size is not altered in hypertrophic obesity, there is an accumulation of senescence-associated β-galactosidase among subcutaneous SVF cells [15]. These observations aside, aberrant fate determination is regarded as a hallmark of stem cell aging, leading to a reduced production of committed progeny [16]. Along with nutrient overload, the stem cell niche becomes hyperinflammatory and unfavorable to their functionality. For instance, the proinflammatory cytokines, TNF and IL1β, which are enriched in the obese adiponiche, restrict the fate specification of ASC towards adipogenic lineage, thereby deteriorating AT homeostasis [41]. Beyond AT, defects in resident stem cell aging are often reported across different tissues during obesity, such as decreased hippocampal neurogenesis, aberrant marrow adipogenesis, as well as enhanced myelopoiesis [42].

Given the reported multipotency of ASCs, an irreversible fate determination towards adipocyte lineage is necessary upon both intrinsic and extrinsic regulations. Regardless of the signaling pathways involved, the mechanisms that promote preadipocyte commitment favor the acquisition of adipocytic characteristics, especially the increased expression of pro-adipogenic key master transcriptional factors, PPARγ and C/EBP [43,44]. While obesity can be characterized by hypertrophy or hyperplasia along with calorie overload, the former type is more predominant in adulthood, and generally considered as metabolically unhealthy, in comparison with hyperplasic obesity [45]. In fact, the senescent ASCs from obese individuals are accompanied by decreased PPARγ expression following adipogenic induction, implying the relationship between ASC aging and lineage commitment to progeny [39]. Inadequate adipogenesis, coupled with increased lipolysis, contribute to adipocyte hypertrophy; such an expansion not only potentiates the tissue inflammation, but also drives the ectopic lipid deposit, ultimately resulting in insulin resistance, systemic inflammation, as well as adiposopathy. Hence, dissecting the regulation of ASC towards adipogenic lineage will be critical in understanding its pathological role in obesity, and help in exploring potential therapeutic target(s) to rejuvenate their stemness.

## 5. Pathological Alternations in Extracellular Signaling Pathways

How ASCs are specified into adipogenic lineage requires a balance between pro-adipogenic and anti-adipogenic signals. When ASCs are dysregulated, the inhibitory stimuli repress the preadipocyte commitment, and also preferentially promote differentiations such as osteoblastogenesis or chondrogenesis, simultaneously. While the crosstalk between BMP, Wnt, and Hh signaling pathways are extensively studied, the pathology of others in association with hypertrophic obesity warrants investigations (Figure 2).

### 5.1. An Adipogenesis Initiator–Bone Morphogenic Protein Signaling Pathway

One of the most characterized signaling pathways is BMP signaling, despite its dual commitment effects on adipogenesis in accordance with the ligand types. BMP2 and BMP4 are the most studied pro-adipogenic factors which trigger both white and brown adipogenesis [46,47]. Through engaging with its receptor BMPR1/2 complex, the SMAD signaling cascade is elicited and hence transduces the signal to nucleus via recruitment of SMAD4 (Figure 2) [48]. The BMP/SMAD signaling cascade is further amplified through the interaction between SMAD complex and zinc-finger protein ZFP423, driving the preadipocyte commitment [49]. Meanwhile, activation of BMPR also induces the non-canonical pathway involving p38 MAPK. Together, these SMAD-dependent and independent pathways not only promote the expression of PPARγ and C/EBP but also promote the expression of cytoskeleton-associated proteins, including lysyl oxidase, translationally-controlled tumor protein-1, and αB crystallin involved in adipogenic commitment [46,50].

Imbalance in commitment programming indeed leads to uncontrolled AT expansion as a root of hypertrophic obesity, providing that BMP4 signaling is negatively regulated by the direct binding between BMP4 and their physiological inhibitors, such as noggin and gremlin. Genetic ablation of noggin in mouse preadipocyte and adipocyte reinforces the BMP4 signaling, resulting in uncontrolled adipogenesis as well as obesity [51]. However, the pathophysiological relevance—whether such deficiency results in adiposopathy in the general population—is being questioned; in particular, mutation in BMP signaling is rarely linked to the pathogenesis of obesity. In fact, along with increasing adiposity, elevated levels of noggin and gremlin were observed in both basic research and clinical observation, subsequently attenuating cellular responsiveness to BMP stimuli known as BMP4 resistance [52]. Similarly, gremlin-2 expression in ASCs is increased upon age-related obesity; as such, it disturbs their adipogenic switch [53,54,55]. The attenuated BMP4 signaling not only blunts the commitment program in ASCs, but also represses the adipocytic lipogenesis which further promotes ectopic lipid storage in muscle and liver, explaining why systemic inflammation occurs in obesity [56].

In addition, sharing the same superfamily and similar molecular characteristics as BMP, TGFβ as a pleiotropic cytokine inhibits the adipogenesis but stimulates the osteochondrogenic gene profile [57]. Activation of TGFBR by its ligand phosphorylates SMAD2/3 complex, in turn represses the C/EBP transactivation function (Figure 2) [58]. High level of TGFβ in blood and ATs was confined to obese individuals; it results in defective adipogenesis which promotes the development of adiposopathy [59]. Together, defective BMP signaling is thought to disrupt the adipogenic switch as a feature of ASC aging upon obesity.

### 5.2. A Regulator towards Wrong Fate–Wingless (Wnt) Signaling Pathway

As a canonical transduction pathway in regulating fate determination during development, the Wnt activation suppresses the preadipocyte commitment, and instead promotes osteogenic or myogenic differentiation [60,61]. Along the adipogenesis, the endogenous expression of these Wnt ligands is downregulated while the negative regulators of Wnt signaling are mutually upregulated, thereby guiding the precursor towards the adipogenic lineage [62,63]. Mechanistically, binding of Wnts to Fzd receptor and LRP5/6 co-receptor inhibits the GSK3β, and in turn enables the translocation of stabilized β-catenin into the nucleus where it serves as transcriptional co-activator with TCF/LEF (Figure 2) [64]. By contrast, when Wnt ligands are absent, GSK3β will promote the ubiquitin-mediated degradation of β-catenin by phosphorylating. In line with this concept, stabilization of β-catenin by other mediators may also interrupt the adipogenic switch, for instance RSPO2/3, the agonists of LGR4/5/6 receptors [65,66]. Previous studies have demonstrated that activation by Wnt3a/6/10a/10b promotes the osteoblastogenesis by enriching its lineage gene profile such as *Runx2*, *Dlx5,* and *Osx* [67,68,69,70]. Of note, it not only blocks the adipocyte commitment, but also disrupts the adipogenic differentiation by downregulating the expression of PPARγ and C/EBP proteins.

In fact, how the anti-adipogenic Wnt/β-catenin pathway interacts with pro-adipogenic BMP signaling is particularly important to lineage specification. WISP2, an adipokine enriched in ASCs, inhibits them from entering preadipocyte commitment through Wnt/β-catenin activation (Figure 2) [69]. Hammarstedt et al. further highlighted the association between Wnt and BMP4 signaling, which address why activating Wnt will lead to decreased PPARγ expression [69]. While the transcriptional activity of ZFP423 is positively regulated by BMP4 activation, WISP2 is intrinsically bound with ZFP423 in cytosol which prevents its nuclear translocation and hence pro-adipogenic transcription.

Hypertrophic obesity is often associated with failure in adipogenesis as a consequence of increased Wnt activation, leading to metaflammation and its complications. Individuals with gain-of-function mutation in LRP5/6 reveal an increased risk of becoming obese via hampering of the adipocyte turnover [71]. Among the nondiabetic obese individuals, WISP2 overproduction was also observed in accordance with adipocyte size positively, thereby reinforcing its inhibitory effect on BMP signaling [69]. Moreover, providing that RSPO2 is a positive regulator of Wnt signaling, a recent study conducted by Wolfrum and his team identified the enrichment of RSPO2 in a particular progenitor subset called Lin^−^ Sca1^+^ CD142^+^ adipogenesis regulator cells, providing a clearer picture about the negative regulatory loop of adipogenesis [72]. Coupling the clinical observations and basic research, adipose RSPO2 and RSPO3 upregulation is a hallmark of obesity, in which these features inhibit the preadipocyte commitment and hence de novo adipogenesis [72,73].

Furthermore, fate determination can be regulated by the β-catenin-independent pathway, given that its prototypical stimulus, Wnt5a/3a, can interact with the receptor complex composed of Fzd and alternative tyrosine kinases such as ROR and RYK [64]. These non-canonical signaling transductions rely on the planar cell polarity pathway via Rho-GTPase; or intracellular calcium release through CaMKKII and/or PKC, thereby modulating the gene expression (Figure 2). In fact, increased Wnt5a expression is confined with visceral obesity, in particular the adipose hypertrophy stimulates Hippo pathway that promotes its expression within the adiponiche [74]. The Wnt5a also stimulates the ROCK activity rather than β-catenin among the ASC population, and in turn promotes the osteogenic specification, as observed by enhanced alkaline phosphatase, activated and upregulated expression of osteogenic genes *Runx* and *Osx* [75]. Together with the proinflammatory role of Wnt5a, enhanced Wnt events contribute to the development of hypertrophic obesity [76].

### 5.3. An Anti-Adipogenic Player at Primary Cilium–Hedgehog (Hh) Signaling Pathway

In response to environmental cues, primary cilium, comprising microtubule-made axoneme and basal body, necessarily helps cells to initiate signal transduction upon stimulation [77]. Being located in this sensory organelle, Hh signaling is a conserved pathway in regulating the organogenesis in both invertebrates and vertebrates. An earlier study has demonstrated Hh stimulation negatively regulates the adipogenesis in both invertebrate and rodent [78]. Mechanistically, the regulation of Hh signaling is governed by two transmembrane receptors, Patched (PTCH1 and PTCH2) and Smoothened (SMO). In mammals, there are three reported Hh ligands, Sonic (SHh), Indian (IHh), and Desert (DHh), which engage with PTCH1 and PTCH2; whereas SMO is an orphan receptor with 7-transmembrane domains [79]. In the absence of Hh stimuli, receptor PTCH inhibits the SMO activation from entering the primary cilium; as such, the transcriptional factor GLI proteins are thereafter phosphorylated and degraded. Therefore, the effect of Hh signaling is silenced. On the other hand, upon the Hh ligand binding, the inhibitory effect on SMO receptor is relieved. As a result, it promotes the activation of GLI proteins from SUFU complex (Figure 2). Molecularly, GLI proteins composed of a classical DNA-binding zinc-finger domain, in particular GLI1 and GLI2, serve as a transcriptional activator while GLI3 is a repressor.

James and his colleagues revealed that SHh-stimulated ASCs preferentially differentiated into osteogenic fate with RUNX upregulation and matrix mineralization, rather than adipogenic lineage [80]. Similarly, on the basis of SMO activation as the prerequisite of GLI transduction cascade, pharmacological agonism by small molecules such as purmophamine or oxysterol also alter adipogenesis but favor the osteogenesis in human MSCs [81,82]. Despite a known anti-adipogenic effect, the underlying mechanism of Hh signaling has remained unsolved for many years. In fact, the interplay between Hh signaling and Wnt signaling in adipogenesis is indispensable, by which GLI2 protein induces the Wnt6 expression, thereby repressing adipogenic responses [83]. Furthermore, Shi and Long proposed that the GLI2-dependent pathway is mediated by Wnt5, exerting its anti-adipogenic and anti-lipogenic roles [84]. Beyond this GLI-dependent pathway, transduction of Hh signaling also links with non-canonical pathways, such as inducing Warburg-like metabolic reprogramming through Ca^2+^ -AMPK axis, and cytoskeletal remodeling via RhoA axis; however, the preferential effect of ligands in choosing the pathways remains unclear.

Genetic ablation of GLI2 or SMO in mice implies defective Hh signaling pathway that results in obesity, but, physiologically, whether dysregulated Hh signaling contributes to hypertrophic obesity warrants further investigation, in particular the missing link with ASCs and preadipocyte commitment [83]. During obesity, the subcutaneous SHh level was observed while IHh and DHh level remained unchanged [84]. Intriguingly, the increased ligand expression does not overactivate GLI-dependent cascade, and instead triggers the MAPK-dependent non-canonical pathway in adipocytes, resulting in PPARγ disability and insulin resistance [84]. Considering the importance of PPARγ in adipogenesis, it is possible that the altered SHh signaling may also switch the commitment away from adipogenic lineage by promoting the phosphorylation and ubiquitination of PPARγ in the adipocyte precursors.

While Hh signaling is closely associated with primary cilium, emerging evidence has suggested the intrinsic correlation between ciliopathies and obesity [85]. Clinical observations suggested that patients with ciliopathies such as in Bardet-Biedl syndrome often become obese and diabetic [85]. Of note, cilia length is an important factor of the adipogenesis in ASCs. Along with adipogenic differentiation, cilia elongation is mediated with increasing recruitment of a pro-adipogenic receptor IGF1R onto the cilium (IGF1R signaling will be further discussed in Section 5.5) [86]. Nonetheless, the cilia length is greatly reduced in ASCs derived from obese individuals because of the proinflammatory factors such as TNF and IL6 (lean: 4.43 ± 0.91 μm vs. obese: 2.76 ± 0.94 μm), resulting in downregulated dynamicity and responsiveness [87]. Such differences within 2 μm on cilia further disrupt the differentiation capability of ASCs, neither adipogenesis nor osteogenesis. These findings provided a mechanistic explanation for why ciliopathies are associated with metabolic syndrome in the context of adipogenesis.

### 5.4. A Contact-Dependent Inhibition–Notch Signaling Pathway

Notch signaling pathway ubiquitously regulates proliferation and differentiation among mammals. Increased Notch activity in stem cells is associated with their aging process, which can be revealed in both obesity and Hutchinson–Gilford progeria syndrome (a premature aging disease) [88,89]. The Notch signaling requires a cell–cell dependent communication, comprising transmembrane NOTCH receptor, and membrane delta-like ligand (DLL) or serrate-like ligand (Jagged; JAG) on neighboring cells [90]. Most of the studies agree that the Notch activation inhibits the adipogenesis in ASCs, as examined by the expression of adipogenic markers [91,92,93,94]. Upon the ligand binding, the receptor is cleaved by γ-secretase, releasing the Notch intracellular domain (NICD) as a transcriptional coactivator with CSL protein to initiate gene expression such as *Sox9*, *Hey1,* and *Hes1*. SOX9 is a well-known transcriptional regulator in tissue development; it activates the expression of MEIS1 which binds onto the promoter region of gene *Cebpd* (which encodes C/EBPδ) and the coding region of gene *Pparg*, as such downregulation of these markers represses the adipogenic differentiation [95]. Similarly, a study on porcine MSCs suggested HES1, another Notch-target downstream molecule, inhibits the adipogenesis via transcriptional repression of FAD24 [96].

In obesity, enhanced Notch signaling is linked to inflammation in perivascular ATs which worsens the vasculature, particularly in relationship with adipogenesis [97]. As shown in the latest report, an upregulation of NOTCH3 receptor was identified in replicative-senescent ASCs derived from obese patients [98]. Such Notch activation by DLL4 and JAG ligands induces the expression of gene *Acta2*, *Col1a1,* and *Inhba*, and in turn shifts the progenitor cell fate from adipogenesis towards myofibrogenesis and senescence-related phenotypes. As a result, the pathological activation of Notch signaling favors hypertrophic obesity.

Of note, the activity of NOTCH receptors is inhibited by their non-canonical ligands, DLK1 or DLK2, which lack a conserved domain for receptor interaction [90]. DLK1 is also known as preadipocyte factor-1 given to its source. Indeed, silencing the Notch signaling by pharmacological or genetic means can enhance the adipogenic capability of ASCs; however, the binding of DLK1 also reveals an anti-adipogenic effect as Notch signaling [94,99]. DLK1 interacts with C-terminal domain of fibronectin, which in turn activates the integrin and its downstream signaling cascade MAPK pathway. Coincidently, such activation also drives the upregulation of SOX9 which inhibits the adipocyte differentiation [100]. Given the positive correlation between body adiposity and DLK1 expression, its pathological role in contributing to hypertrophic obesity has been proposed [101].

### 5.5. A Stimulator who Declines over Time—Insulin-like Growth Factor Signaling Pathway

IGF is an evolutionally conserved peptide known to regulate cellular growth in accordance with nutrient availability. Most of the studies have reported the pro-adipogenic effect of IGF1 (formerly known as somatomedin C), rather than IGF2. When IGF1 is bound with its cognate receptor IGF1R, it elicits the MAPK cascade which antagonizes the Wnt/β-catenin pathways via Axin2 upregulation, which in turn triggers the gene expression of PPARγ and C/EBP [102]. Therefore, genetic ablation of IGF1R will result in exceedingly reduced adipose mass [103]. Furthermore, the activation of AKT is thought to involve following the ligand engagement on IGF1R, in which the phosphorylated AKT suppresses the activity of FOXO1 which is known as an adipogenesis inhibitor [104,105].

Regarding the structural homology of IGF with insulin, both IGF1 and IGF2 can interact with insulin receptor, and vice versa. In particular, this ensures the continuity in physiological function when one of them is inactivated. While IGF1 can activate the downstream cascade of insulin receptor as an alternative method of promoting adipogenesis, IGF2 also regulates the adipogenic specification through engaging with IGF1R and insulin receptor, beyond its cognate receptor IGF2R [106,107].

Moreover, IGF is a hormone associated with cellular aging. Given that IGF is secreted from the liver upon the stimuli of growth hormone, its serum level declines after puberty and remains very low level after 60 years old [108]. The phenomenon of reduction of growth hormone and IGF is known as somatopause. While it is thought to be a driver of cellular aging, low circulating IGF level is often reported in obese individuals, implying that obesity’s result in cellular aging is likely attributable to obesogenic somatopause [109]. As such, we hereby propose that decreased bioavailability of IGF in obesity not only promotes the ASC aging phenotype such as stress accumulation, but also disrupts the adipogenic switch that favors the progression of hypertrophic obesity.

In summary, pathological alternations in these signaling molecules interrupt the lineage determination of ASCs, and it defines a hallmark of cellular aging upon obesity. Most importantly, commitment dysfunctions in ASCs restrict AT expandability as a root of hypertrophic obesity, but it nevertheless offers a therapeutic strategy to guide ASCs back to adipogenic commitment [52].

## 6. Epigenetic Drift

Epigenetic modification necessarily guides the ASCs towards adipogenesis through regulating the gene transcription in relation to adipogenic markers or abovementioned signaling molecules. How cellular aging or senescence emerge can be explained by defective maintenance in the epigenome as a cause of epigenetic drift [16]. In the majority research on preadipocytes, the epigenetic events underlying fate determination in ASCs are limited; in particular, some regulators such as lncRNA ADIPINT are only expressed in preadipocytes [110]. More profoundly, considering the obese adiponiche is rich in cellular stress, our understanding on how obesity influences ASC epigenome and fate dysregulation is yet to be thoroughly investigated (Table 2).

DNA methylation involving the 5-methylcytosine or N6-methyladenine (m6A) is a way to silence the gene expression through either recruiting transcriptional repressor or inhibiting the binding of transcriptional activator. Given that ZMAT3 is regarded as a senescence-related marker that activates p53, its DNA hypomethylation in adipose precursor favors senescence in close relatives of diabetic patients, thereby disrupting adipogenesis [111]. Despite the unclear mechanism in altering the methylation pattern, these findings also suggested the predisposition of metabolic syndrome in certain individuals with family history. Similarly, methylation in RNA level by is shown in regulating their stability and trafficking. IMP2 (or known as IGF2BP2) is a reader of N6-methyladenosine to regulate mRNA stability and translation, despite the fact its intronic SNP is well known in association with genetic predisposition of type-2 diabetes [112,113]. A latest study published in *Diabetes* revealed IMP2 is nonredundant in preadipocyte commitment in ASCs [114]. Molecularly, the engagement between IMP2 and Wnt receptor Fzd8 mRNA promotes the latter degradation through recruitment of CCR4-NOT deadenylase under the presence of mTOR.

**Table 2 cells-12-00662-t002:** Recent discoveries in epigenetic modification of adipogenesis in ASC.

Types	Proteins/RNA	Functions in ASC	Effect on Adipogenesis	Reference
DNA modification	ALKBH1	Demethylates 6mA of *Gys1* and *Hif1a*	+	[115]
	TET3	Erases DNA methylation at CEBP binding motif	+	[116]
Histone modification	PRMT1	Catalyzes H4R3me2a at gene *Pparg*	+	[117]
	SIRT1	Deacetylates gene *Sfrp* for activating Wnt signaling	-	[118]
	SIRT2	Deacetylates gene *Foxo1* for inhibiting *Pparg* expression	-	[118]
RNA modification	IMP2/IGF2BP2	Promotes degradation of *Fzd8* mRNA	+	[114]
miRNA regulation	miR-27a-3p	Targets *Pparg* mRNA	-	[119]
	miR-138	Target the mRNA for lipoprotein lipase	-	[120]
	miR-424(322)/503 cluster	Targets γ-synuclein that regulates lipid metabolism	-	[121]
	miR-503-3p	Targets *Wnt2* and *Wnt7b*	+	[122]
lncRNA regulation	HOTAIR	Facilitates the actin reorganization and lipogenesis	+	[123]
	LYPLAL1-AS1	Modulates the stability of desmoplakin Inhibits Wnt/β-catenin pathway	+	[124]
CircRNA regulation	CircFOXP1	Indirectly regulates Wnt5a expression	+	[125]
	circRNA_23525	Targets miR-30a-3p	-	[126]

Abbreviation: 6mA: N6-methyldeoxyadenosine; ALKBH: Alkb homolog 1, histone h2a dioxygenase; circRNA: circular RNA; GYS: glycogen synthase; HIF: hypoxia-inducible factor; IMP: Insulin-like growth factor 2 mRNA-binding protein 2; lncRNA: long non-coding RNA; miRNA: microRNA; PRMT: protein arginine methyltransferase; SIRT: sirtuin; TET: ten eleven translocation; Wnt: Wingless.

In comparison to DNA methylation, the modification at histone level is widely discussed in terms of adipogenesis. Through enzymatic reaction on particular residues (such as lysine and arginine) at N’ terminal of histone tail, it reversibly regulates the chromatin accessibility for transcriptional factors between euchromatin and heterochromatin in accordance with types of modification, such as methylation as gene silencing and acetylation as gene activation [127]. Previous findings have shown SIRT1 and SIRT2 are known as NAD^+^ -dependent histone deacetylases that are unfavorable to adipogenic commitment, respectively, and their overexpression in obesity further reinforces the inhibitory effect on adipogenesis [118]. It indicates that overnutrition promotes vulnerability in the epigenome, which in turn results in aberrant differentiation and adipose hypertrophy.

Moreover, the non-coding RNA occupying 98% of the genome is versatile, which contributes to developmental complexity [128]. Among the RNA transcripts, microRNA (miRNA) and long non-coding RNA (lncRNA) are extensively studied in the context of adipogenesis. In general, miRNA is bound with 3′ UTR of target gene with strong complementarity, thereby promoting mRNA degradation. Most recently, a cluster of miR-424(322)/503 was found in tacking of the expression of SNCG, which regulates the fatty acid metabolism and adipocyte differentiation in ASCs [121]. As such, during adipogenesis, the miRNA cluster is transcriptionally repressed to enable the adipogenic program. Increased circulating SNCG level was identified in obese individuals while it was reduced in patients following gastric bypass surgery [121]. Despite unknown reason in triggering SNCG expression according to adiposity, SNCG overexpression indeed exacerbates the AT enlargement by simply interrupting the fatty acid metabolism. Unlike miRNA, how lncRNA regulates the cellular behavior involves a variety of mechanisms, including but not limited to chromatin remodeling, splicing, and protein stabilization [128]. Indeed, even though overexpression or knockdown of these regulators may skew the adipogenic switch, their pathological alternation in ASCs remains to be clarified in obesity.

Overall, while cellular aging can be explained by a concept of an “epigenetic clock”, how these regulations accelerate the ASC dysregulation in the obese adiponiche has been queried for many years. It is noted that the epigenetic modification is not limited to the abovementioned routes; it will be interesting to study others such as chromatin remodeling, PIWI-interacting RNA (piRNA), and small nucleolar RNA (snoRNA) to complete the picture of adipogenesis [128].

## 7. Mechanical Stretching

Sensing the mechanical cue is nonredundant in regulating adipogenesis, given a limited space among the adiponiche. During obesity, AT expansion is observed in both subcutaneous and visceral depots, in which the mean cell volume of adipocyte is increased by 6-8-fold in obesity [129]. Undoubtedly, the pathological AT growth exerts stress on neighboring cells. The extracellular matrix (ECM) is crucial to the tissue architecture, especially to AT dynamicity. AT is supported by the network rich in collagen fibers and fibronectin. It not only provides an attachment to support the host cells, but also transduces the mechanic stimuli via membrane integrins anchored onto ECM. Along with AT expansion, integrins as gatekeepers suppress the adipogenesis, thereby restraining the AT growth. Integrins are the transmembrane receptor assembled from alpha and beta subunits, variants in subunit by differential RNA splicing result in 24 heterodimeric combinations that determine the ligand specificity, such as α_1_β_1_ for collagen, α_6_β_4_ for laminin, as well as α_v_β_1_ for recognizing Arg-Gly-Asp motif as seen in fibronectin [130]. Importantly, the hypertrophic growth of adipocyte is always accompanied with ECM remodeling as shown by increased myofibroblast and fibrotic structure, such that the integrin signaling is reinforced in obese adiponiche surrounding with mechanical stretching. Following the ligand binding on integrins, the reassembly of intracellular microtubule triggers several intracellular signaling cascades. For instance, the RhoA-ROCK pathway is activated which promotes the F-actin polymerization and stress fiber formation, such that the ASCs become spreading in shape, in turn differentiate towards osteogenic lineage [131,132].

Furthermore, the mechanical stretching activates another mechanosensitive pathway called Hippo pathway which is composed of two co-transcriptional activators, YAP and TAZ. Following stress-induced cytoskeleton reorganization, the translocation of YAP and TAZ from cytosol to nucleus enables them to partner with other transcriptional factors, thereby modulating the gene expression [133]. Previous findings have confirmed TAZ drives the osteogenesis, as a result of repressed PPARγ expression and upregulated RUNX2 expression [134]. Similarly, YAP also plays a role in promoting osteogenic differentiation via stabilizing the anti-adipogenic effector β-catenin. Together, while most of the research focus on the how mechanical stress influences ASCs in the context of regenerative medicine, mechanobiology about ASCs in obese adiponiche remains a pivotal question; in particular, emerging studies have highlighted its role in stem cell commitment.

## 8. Metabolic Reprogramming

Stem cell maintenance is interdependent with a network of metabolic pathways in response to both intrinsic and extrinsic stimuli, such as stemness and nutrient availability. For example, inhibiting the eicosanoid signaling pathway in embryonic stem cells not only maintains high level of ω-6 and ω-6 fatty acid in embryonic stem cells, but also enhances their pluripotency by delaying the loss of OCT4 and NANOG [135]. Notably, fate determination can be sequential that requires metabolic reprogramming prior to transcriptional transitions [136].

While the majority of the research focuses on how the oxidative stress causes cellular damages, low level of ROS generated from mitochondrial complex III is indeed essential to adipogenic differentiation in the mTORC1-dependent regulation [137]. Interestingly, antioxidant reduces the adipogenic potential of stem cells, but it can be rescued by hydrogen peroxide. Given the unique microenvironment in ATs, ASCs survive under a hypoxic condition around 1–5% oxygen content. Although hypoxia is also closely associated with ROS generation, it simultaneously favors the adipogenic differentiation, by which it may enhances the glucose uptake and metabolism via HIF1α [138,139].

Furthermore, the metabolites derived from Krebs cycle are also necessary to regulate fate determination. Playing a central role in orchestrating cellular metabolism, NAD^+^ is a coenzyme involved in a variety of redox reactions. As NAD ^+^ cannot pass through cell membrane, its de novo biosynthesis is carried out from (1) L-tryptophan through kynurenine pathway; (2) vitamin precursor such as nicotinic acid through Preiss–Handler pathway; and (3) savage pathway involving SIRT [140]. Notably, the NAD ^+^ bioavailability is essential to adipogenic program in human MSC, in which increased level of NAD ^+^ or supplementation of its percussors β-nicotinamide disrupt adipogenesis as well as lipid accumulation [140]. However, excessive NAD^+^ bioavailability not only increases stress responses, but also hinders the oxidative phosphorylation and subcellular compartmentalization of NAD ^+^ metabolism during adipogenesis, resulting in downregulated adipogenesis.

Moreover, α-ketoglutarate, a derivative in Krebs cycle and component for amino acid, involves in DNA methylation as a co-factor by TET enzymes. Given that TET3 is a pro-adipogenic protein, dietary supplementation of α-ketoglutarate was found to prevent metabolic syndrome in mice [116,141]. These findings also suggest the potential of dietary intervention against metabolic syndrome, in which enhanced adipogenesis in WAT may contribute to the beneficial effect of dietary supplementation.

In addition, ketone bodies are an alternative energy source during starvation. Through breaking own fatty acids, ketogenesis refers to the production of acetone, acetoacetate, and β-hydroxybutyrate (BHB). In beige adipocyte, the biosynthesis and export of BHB are promoted by beige marker PRDM16 [142]. As a paracrine factor, BHB shapes the beige progenitor and subcutaneous ASCs away from myofibrogenesis, but towards adipogenesis. Mechanistically, although BHB may directly inhibit some histone deacetylases, its catabolism via β-hydroxybutyrate dehydrogenase is required to the pro-adipogenic effects, suggesting that acetoacetyl-CoA, acetyl-CoA, and succinyl-CoA may modulate the adipocyte commitment.

## 9. Emerging Therapeutics

Considering that the historical view indicates environmental factors as a culprit of obesity, the approved weight-loss medications nowadays are on the basis of restriction in calorie intake. For instance, orlistat is a pancreatic lipase inhibitor which hampers the fat absorption in the gut; others include sympathomimetics (e.g., phendimetrazine), GLP-1R agonist (e.g., liraglutide), and dopamine-and-noradrenaline reuptake inhibitor (i.e., Naltrexone bupropion) which suppress the appetite through their corresponding neuroendocrine functions [143]. Likewise, the majority of drug candidates under ongoing clinical trials also target the endocrine control of food intake, particularly leptin signaling, neuropeptide Y signaling, amylin signaling, as well as ghrelin signaling. Remarkably, perhaps, failure in drug development is very common due to pharmacological difference between animals and humans, the safety concerns from neuroendocrine aspects indeed account for the obstacle in pharmaceuticals against obesity, particularly uncertain cardiovascular safety and potential in undernutrition [143]. Considering the high comorbidity of obesity with cardiovascular diseases, sympathomimetic itself may worsen the cardiovascular events through interfering with the vascular tone.

In recent years, advances in pathological understanding of obesity coupled with the AT expansion unquestionably allow us to revisit its etiology from alternative perspectives, particularly the interplay between obese milieu and ASC dysfunction. As such, reversing stem cell aging may be promising against the progression of obesity and related complications. Metformin as a glucose-lowering drug improves insulin sensitivity and alleviates cellular stress responses in diabetes [144]. Findings revealed the use of metformin not only reduces the unhealthy AT expansion in obese individuals, but also improves the stemness and prevents senescent phenotypes in ASCs [145]. Similarly, rosiglitazone is an anti-diabetic thiazolidinedione that stimulates the adipogenic differentiation, in turn improves insulin sensitivity and reduces metaflammation [146]. Similar observations on a cohort with obesity suggested another thiazolidinedione, pioglitazone, also promotes the adipocyte turnover that improves their metabolic health [147].

In addition, the abovementioned studies have shed light on how aberrant fate determination in ASCs results in hypertrophic obesity, and whether guiding the stem cells’ back adipogenic lineage will be a strategy against adipocyte hypertrophy. In fact, findings from basic research support this concept as addressing the root of obesity (Table 3). For example, while the BMP resistance, by which the BMPR responsiveness is attenuated and BMP inhibitors are highly expressed, is a feature in obesity, recombinant BMP or BMP overexpression may serve as promising treatment against adiposopathy. Nonetheless, the current research in relation to targeting these dysfunctional pathways in obesity does not gain much research attention as anti-obesity treatment; exploring the feasible drug candidate(s) against ASC aging will be encouraged.

Moreover, senolytic and senomorphic are also popular strategies in restoring tissue homeostasis in relation to cellular senescence. Considering the interplay between obese adiponiche and cellular senescence, hampering the negative effects brough from proinflammatory and cytotoxic secretome of senescent cells offers a novel avenue of treating obesity. Senolytic drug is developed to eliminate the senescent cells from the niche by promoting their death. Cocktail of dasatinib (a tyrosine kinase inhibitor) and quercetin (a natural flavonoid) attenuated the AT inflammation and related complications brought by obesity [155]. Moreover, senomorphic drug offers alternative therapeutic strategy by suppressing the detrimental effects of senescence without promoting senescent cell death. Currently, many FDA-approved drugs have been reported as senomorphic class, such as metformin, aspirin, and statin, while many other small molecules have been developed in reversing SASP [156]. In particular, obesogenic senescence is not limited in AT, but also other tissues such as brain tissue. Further research will be worthy in clarifying their efficacy as well as any potential side effects; these treatments will be promising in the foreseeable future.

## 10. Perspectives

Stem cell aging is not only characterised by stemness or renewal capability: whether they can differentiate into the correct lineage in their corresponding host tissues is also a crucial consideration [16]. While multiple regulators of the lineage commitment have been shown to become dysregulated in obesity, there are other possible mechanisms which link ASC dysfunction and hypertrophic obesity together. The peripheral circadian clock is an auto-regulatory loop regulating cellular behaviors via oscillatory gene expression machinery. Previous study has shown how core clock proteins, PER3 and BMAL1, positively regulate the adipogenic differentiation via KLF14, demonstrating the link between molecular clock and lineage commitment [157]. Importantly, findings from obese patients indicated a clock disruption in both omental adipocytes and precursors, in which such alternation is closely associated with the proinflammatory gene profile [158]. On the basis of proinflammatory cytokines in inhibiting adipogenesis, obesogenic clock dysregulation is likely to skew away the adipogenic differentiation, though further experiments are required [41].

Another interesting mechanism raised is whether autophagy plays a role in lineage commitment, given that it is closely associated with cellular metabolism. As a conserved process, it is a degradative mechanism that catabolizes unnecessary cellular components upon stress stimuli [159]. During obesity, activated mTORC1 is thought to suppress autophagy, resulting in dysregulation in eliminating toxic components. A recent report demonstrated that reduced microautophagic activity by AFF4 depletion represses the adipogenesis [160]. Furthermore, despite evidence on preadipocyte line 3T3-L1, chaperone-mediated autophagy is activated along with adipogenesis, whereas its dysregulation impairs the preadipocytes from entering differentiation [161]. Although further studies are required to dissect their interplay, it can be hypothesized that decreased autophagic flux in ASCs in the obese adiponiche will contribute to adipose hypertrophy. As such, pharmacological intervention into autophagy may be able to restore the adipogenic capability in dysregulated ASCs.

The bidirectional crosstalk between ASC and leukocyte should not be overlooked, particularly that the obese adiponiche is infiltrated with proinflammatory leukocytes. For instance, IL33 secreted from ASC promotes the activation of ILC2s and Treg, which in turn maintain tissue homeostasis [162]. Aside from the paracrine factors, cellular reprogramming can be achieved by direct cell–cell communication. Membrane expression of PDL1 and galectin-9 is considered to be the immune checkpoint that interacts with effector T-cells [163]. Notably, obesity is thought of as one of the contributing factors to cancer. Immunotherapy blocking PDL1 and PD1 by monoclonal antibodies is a potent treatment in various types of cancer through boosting the immunogenic responses [164]. Nonetheless, the overactivation of cytotoxic cells creates a safety concern in immunotherapy as several complications have been reported. For instance, findings from clinical data suggested the use of PDL1 or PD1 inhibitors demonstrates a substantial risk in developing type-2 diabetes as well as cardiovascular events [165]. These observations raise the question in adipose biology of whether these blockades also activate the leukocyte population in interrupting ASC homeostasis, and hence exacerbate adiposopathy and related complications [164,166]. If it happens, pharmacological intervention should be employed together with these anti-cancer immunotherapies. Collectively, these missing immunomodulatory effects in ASC and neighboring leukocytes among the niche await further investigation.

Last, the research on promoting adipogenesis as anti-obesity treatment is promising but limited (Table 3). Notably, the pace of a new drug from bench to actual clinical practice takes time, because of molecular optimization, incompatible results between basic research and human trials, enormous costs, multiple rounds of clinical trials, as well as safety concerns [143,167]. In this regard, drug repositioning is a strategy in searching for feasible drug candidates against common and rare diseases; in particular, the existing FDA-approved drugs have already passed the pharmacokinetics and toxicological examination in human. While targeting the abovementioned signaling pathways is a strategy in treating other diseases like cancer clinically, it is thus important to ask whether these existing drugs may restore the adipocyte turnover and improve metabolic health. For instance, as mentioned in Table 3, pirfenidone counteracts the TGFβ bioavailability in patients with idiopathic pulmonary fibrosis, while vismodegib is the inhibitor for Hh signaling used in cancer treatment [84,151]. It should be noted that further research on animal models will be imperative in examining their anti-obesity potential and underlying safety considerations.

## 11. Conclusions

In summary, it is evident that deficit in stem cell functionality is a hallmark of obesity. As a consequence of failure in adipogenesis, adipocyte hypertrophy is a major culprit of metaflammation and other complications. In particular, the interpretation regarding microenvironmental control of ASCs is critical to our knowledge in etiology of obesity as well as discovery of next-generation drugs. While our understanding of the interplay between ASC aging and adipose hypertrophy is increasing, there is still an unsolved fundamental question: which came first, the chicken or the egg?

## Figures and Tables

**Figure 1 cells-12-00662-f001:**
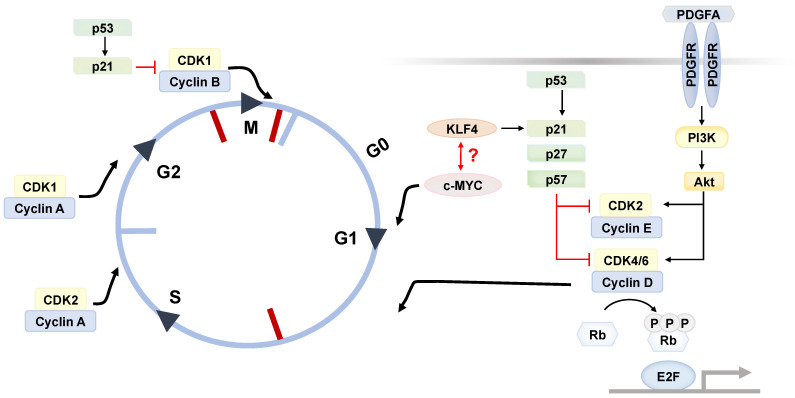
Schematic summary of cell cycle regulation in ASCs. The mammalian cell cycle is the process composed of four phases: growth 1 (G_1_), DNA synthesis (S), growth 2 (G_2_), as well as mitosis (M). Similar to other adult stem cells, it is often that ASC enters the quiescent state (G_0_) when there are no mitogenic stimuli. To ensure the cellular quality, the cell cycle progression is controlled by three major checkpoints, including G_1_/S checkpoint, the G_2_/M checkpoint, and metaphase checkpoint (as indicated by red-colored bar). Mechanistically, these checkpoints are governed by the activity of cyclin and its partner cyclin-dependent kinase. For instance, the complex composed of E-type cyclin (Cyclin E) with cyclin-dependent kinase 2 (CDK2) inactivates the retinoblastoma protein (Rb) via phosphorylation (P), and in turn relieves the inhibitory effect on E2F transcription factor. As such, this enables ASC to undergo gene profile reprogramming for entering S phase. By contrast, the cell cycle progression can be arrested by CDK inhibitors via blocking the binding between cyclin and corresponding CDK.

**Figure 2 cells-12-00662-f002:**
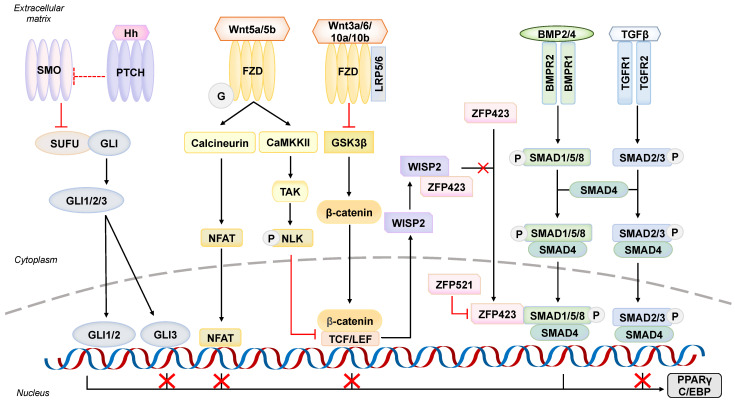
Schematic diagram on BMP, Wnt, and Hh signalling. The mechanism of lineage commitment in ASCs is extensively studied in relation to BMP, Wnt, and Hh signaling. It should be noted that, while BMP signaling pathway is pro-adipogenic and promotes the expression of PPARγ and C/EBP, the TGFβ, Wnt, and Hh signaling pathways are anti-adipogenic (as marked with red cross). As observed in both rodents and obese patients, dysregulation in these pathways restricts the AT expandability and hence promotes the adipocyte hypertrophy. Nevertheless, this also offers a variety of potential therapeutic targets against obesity, by which restoring the adipogenic capacity can maintain the metabolic health and adipocyte turnover.

**Table 1 cells-12-00662-t001:** Overview of adipocyte physiology.

		White Adipocyte	Beige Adipocyte	Brown Adipocyte
Development origin		Dermomyotome in somite and lateral plate mesoderm (except for craniofacial adipocytes as ectodermal descendant)		
Embryogenic progenitor		PDGFRa ^+^ CD29 ^+^ CD44 ^+^	(1) MYH11, ACTA2 (2) Mature white adipocyte	EN1^+^ PAX7^+^ MYF5^+^
Lineage commitment during embryogenesis		FGF10	(1) Selective expression of BMP7 (2) Unclear	Selective expression of PRDM16 and BMP7
Predominant depots	Human	Omental (visceral) Gluteofemoral (subcutaneous)	Within subcutaneous WAT	Supraclavicular
	Rodent	Gonadal (visceral) Inguinal (subcutaneous)		Interscapular
Phenotypes		Single but large lipid droplet Few mitochondria	Multiple but small lipid droplets Abundant mitochondria	
Molecular patterns		ASC1^+^ UCP1^low^ HOXC9^+^	PRDM16^+^ UCP1^high^ PGC1α^+^	CD40^+^ CD142^+^ UCP1^high^
Major Functions		Energy storage Endocrine function	Thermogenesis Clearance of ectopic lipid	Thermogenesis Anti-inflammation
As a secretory organ		Adiponectin, leptin, adipsin	Prostaglandins, 12,13-diHOME, FGF21, NRG4, IL6	
Changes upon obesity		Hypertrophy Cellular senescence	Whitening (transdifferentiation)	Inflammation Oxidative damage

A comparison between white adipocytes, beige adipocytes, and brown adipocytes in accordance with their characteristics [19,20,21,22]. Of note, there are two differentiation origins of beige adipocytes, (1) de novo adipogenesis and (2) white-to-beige transdifferentiation. Abbreviations: 12-13-diHOME, 12, 13-dihydroxy-9Z-octadecenoic acid; BMP: bone morphogenic protein; ER: endoplasmic reticulum; FGF: fibroblast growth factor; IL: interleukin; NRG: neuregulin; PGC: peroxisome proliferator-activated receptor gamma coactivator; PRDM16: PR domain containing 16; UCP: uncoupling protein.

**Table 3 cells-12-00662-t003:** Highlights in experimental therapeutics against obesity.

Dysfunction in	Treatment	Strategy	Reference
BMP signaling	AAV-BMP4	Ligand overexpression	[148]
	AAV-BMP7	Ligand overexpression	[149]
	Recombinant BMP9	Reinforces stimuli to BMPR	[150]
TGFβ signaling	Anti-TGFβ	Blocks TGFβ activity	[59]
	Pirfenidone	Inhibits TGFβ production	[151]
Wnt signaling	AAV-SFRP5	Inhibits the Wnt ligands	[152]
Hh signaling	Vismodegib	Inhibits SMO activity	[84]
Notch signaling	Dibenzazepine	Inhibits γ-secretase	[153]
	Anti-DLL4	Blocks DLL4 activity	[92]
ROCK signaling	Fasudil	Inhibits ROCK	[154]

Abbreviation: AAV: adeno-associated virus; BMP: bone morphogen protein; BMPR: BMP receptor; DLL: delta-like ligand; PPAR: Peroxisome proliferator-activated receptor; SFRP: secreted Frizzled-related protein; SMO: smoothened; TGF: transforming growth factor; TGFBR: transforming growth factor beta receptor; WAT: white adipose tissue; Wnt: Wingless.

## Data Availability

Not applicable.

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
