# Peer review of "A Wrong Fate Decision in Adipose Stem Cells upon Obesity"

_cells, 2023, doi:10.3390/cells12040662_

Round 1

Reviewer 1 Report

This is a great paper highlighting the factors affecting ASC dysregulation in obesity, and even includes the commonly-ignored effects of the extracellular matrix on disease state. The authors have also done a great job of highlighting some of the emerging therapeutics, and briefly touching upon the need to evaluate ASC-immune cell interaction as a potential cause. 

The authors could have delved slightly deeper into the molecular mechanisms of proteins/growth factors sequestered in the ECM and how these affect adipogenesis and obesity in general.

Line 580-581: It is unclear what the authors mean in this sentence, please rephrase. 

There are some minor typos littered throughout the paper. eg. Line 51. 

Author Response

Thank you for your comments. Attached is the revised manuscript. Please feel free to tell us if there is any problem. 

Reviewer 2 Report

The submitted review entitled “A Wrong Fate Decision in Adipose Stem Cells upon Obesity” written by Yiu Ming Cheung and colleagues describes the overview of adipocyte stem cells (ASC) aberrant differentiation with particular emphasis on the role of the signaling pathways, epigenetic modifications, mechanical stretching, and metabolite reprogramming. It is presented in eleven structured chapters including the introduction, characterization of heterogenic adipocytes physiology, their ASC lineage, and cell cycle regulation, followed by the above-mentioned epigenetic, mechanical, and metabolic aspects then summarized in future perspectives and final considerations. Importantly, in each paragraph, the impact of obesity is considered and discussed.

As obesity is now a global pandemic, the understanding of pathological processes linked to adipose tissue changes, including ACS dysfunction, is very important and may shed more light on possible therapies against obesity, which highlights a need for such reviews. The manuscript is outstanding, very solid, and well-written based on recent research articles and reviews. In my opinion, it can be published in actual form, however, I suggest some modifications listed below.

-          Line 32: More common nomenclature for stromal vascular cells (SVCs) is stromal vascular fraction (SVF) cells.

-          In table 1, it is not clear what the numbers for beige adipocytes mean. Is it referring to different progenitor origins? Please specify.

-          In figure 1, lines leading to CDK2 and CDK4/6 are confusing and suggest that p57 and AKT regulate both CDK2-cyclin E and CDK4/6-cyclin E complexes. Moreover, in line 141 is stated that p57 downregulates the cellular quiescence but the line is arrowheaded. Please verify it.

-     Line 166 and herein: According to the HUGO Gene Nomenclature Committee database, tumor necrosis factor-alpha is now simply called tumor necrosis factor (TNF).

-          Line 192 and herein: More common abbreviation for the Wingless pathway is Wnt instead of WNT.

-          Line 218: Please specify which gremlin molecule is mentioned.

In summary, I listed some shortcomings of the submitted review, however, they do not affect the high level of work. The work is very solid, maybe a little too long, well outlined, and organized. Due to the all above-highlighted aspects, the manuscript in its present form can be recommended for publication in Cells after minor improvements.

Author Response

(The authors gave the same response as above.)

Reviewer 3 Report

The authors focused on several important aspects of ASCs, the most innovative are the section on aging or extracellular signaling pathways and epigenetic drift, but also the adipocyte physiology or metabolic reprogramming are necessary into this review. The work has a significant contribution to the field, is well organized.

Author Response

Thank you for your comments. Attached is the revised manuscript as requested by other reviewers. Please feel free to tell us if there is any problem. 

Reviewer 4 Report

Dear Authors

I've read through the review several times. First of all, this is a review and adipose tissue and its functions in our body, the role of adipose tissue in obesity are explained with mechanisms in this review. In particular, genetic mutations and intracellular pathways are illustrated beautifully and in detail. The subject is long and the general subject is well summarized, even though the mind is distracted by the mechanisms while reading. The tables and figures are beautifully designed, explanatory and very clear. It has been explained in great detail which mechanism causes which problem. Although they mentioned that disorders in stem cell functionality are associated with obesity, they compared this situation to a chicken-egg conversation, since there are still uncertainties about the treatment and mechanism.

Author Response

(The authors gave the same response as above.)
